# Observations of the macrophysical properties of cumulus cloud fields over the tropical western Pacific and their connection to meteorological variables

**Michie Vianca De Vera[1], Larry Di Girolamo[1], Guangyu Zhao[1], Robert M. Rauber[1], Stephen W. Nesbitt[1], and Greg M. McFarquhar[2,3]**

[1]Department of Climate, Meteorology and Atmospheric Sciences,
University of Illinois Urbana-Champaign, Urbana, IL 61801, USA
[2]Cooperative Institute for Severe and High-Impact Weather Research and Operations,
The University of Oklahoma, Norman, OK 73072, USA
[3]School of Meteorology, The University of Oklahoma, Norman, OK 73072, USA

**Correspondence:** Michie Vianca De Vera (mdevera2@illinois.edu)

**Abstract.** The poor representation of the macrophysical properties of shallow oceanic cumuli in climate models contributes to the large uncertainty in cloud feedback. These properties are also difficult to measure because it requires high-resolution satellite imagery that is seldomly collected over ocean. Here, we examine cumulus cloud macrophysical properties, their size, shape, and spatial distributions, over the tropical western Pacific using 170 15 m resolution scenes from Terra's Advanced Spaceborne Thermal Emission and Reflection Radiometer (ASTER) collected during the 2019 Cloud, Aerosol and Monsoon Processes Philippines Experiment (CAMP$^2$Ex) mission. The average cloud fraction (CF) was 0.12, half of which was contributed by clouds less than 1.6 km in area-equivalent diameter. This compared well to Terra's Multi-angle Imaging SpectroRadiometer (MISR) resolution-corrected CF of 0.14 but less than the 0.19 measured by Terra's Moderate Resolution Imaging Spectroradiometer (MODIS). The cloud size distribution exhibited a power law form with an exponent of 2.93 and an area–perimeter power law with a dimension of 1.25. ASTER, MISR, and CAMP$^2$Ex aircraft lidar showed excellent agreement in the cloud top height (CTH) distribution peak altitude of $\sim 750$ m. We examined cumulus properties in relation to meteorological variables and found that the variation in mean CTH is controlled most by the total column water vapor, lower-tropospheric stability (LTS), and estimated inversion strength (EIS). The variation in CF is most controlled by surface wind speed and near-cloud relative humidity instead of LTS/EIS, suggesting the need to improve low-cloud parameterizations in climate models that use LTS/EIS based on stratocumulus studies.

## 1 Introduction

Tropical oceanic low clouds, such as shallow cumulus, have been found to explain most of the inter-model spread in global mean cloud feedback (Bony and Dufresne, 2005; Zelinka et al., 2016). This is primarily due to challenges in realistically simulating these clouds at the subgrid scale of global climate models (Ceppi and Nowack, 2021; Myers et al., 2021). Consequently, large-eddy simulations (LESs) have been utilized to resolve low-cloud processes, such as their response to changes in lapse rate, in surface fluxes, and in surface temperatures, to predict the low-cloud feedback (Rieck et al., 2012; Zhang et al., 2012; Bretherton, 2015). These processes are tightly coupled to cloud macrophysical properties. For example, cloud fraction within a domain strongly modulates radiative effects (Chen et al., 2000; George and Wood, 2010; Bender et al., 2016). The cloud top height and spatial distribution also strongly impact the radia-

tion field (e.g., Chen et al., 2000; Tobin et al., 2012). For example, Tobin et al. (2013) found that the outgoing longwave radiation increases, while the shortwave radiation decreases, as the cloud field becomes more aggregated, showing that the emerging radiation field can depend on the spatial organization and geometry of clouds. Other studies (Rampal and Davies, 2020; Goren et al., 2023; Lang et al., 2024) have further shown how cloud morphology and cloud heterogeneity can impact the measured radiative field. Their results imply that cloud macrophysical properties must be considered in the parameterization of the highly interactive processes of radiative heating, turbulent and convective mixing, and cloud microphysical processes that govern the variability in low clouds in LES models (Klein et al., 2017). To mitigate the uncertainties in model parameterizations, it is crucial to better constrain cloud models using observational data.

To help understand cloud macrophysical properties, long-term data from satellite observations are desirable. Jones et al. (2012) found that satellite imagers with a spatial resolution of 80 m or less are needed to accurately measure cloud fraction to an error of 0.01, instrument and algorithm cloud detection sensitivity aside, in the trade wind regions of the world. Most meteorological satellite imagers, however, have a spatial resolution of around 1 km, which is larger than the typical size of shallow cumulus clouds (e.g., McFarquhar et al., 2004; Zhao and Di Girolamo, 2007). This can result in substantial overestimates in cloud fraction (e.g., Shenk and Salamonson, 1972; Di Girolamo and Davies, 1997; Zhao and Di Girolamo, 2006). For example, Dey et al. (2008) found that when pixel resolution degrades from 15 m to 1 km, the mean cloud fraction can increase 4-fold, while the total number of clouds can reduce 26-fold for trade wind clouds. Given the importance of having high-resolution data, observations of cumulus cloud fields should ideally come from land-based imagers. Such high-resolution (< 80 m) freely available satellite data are found in land-based satellite missions, such as Landsat (Crawford et al., 2023) and Advanced Spaceborne Thermal Emission and Reflection Radiometer (ASTER; Abrams, 2000). But such land-based imagers normally collect very little data over ocean, have radiometric gain settings that may not be appropriate for cloud analyses, and have sampling and archiving strategies that are not conducive to forming statistically unbiased properties of clouds. Therefore, there remains the challenge of acquiring high-resolution data over the oceans for studies of cloud macrophysical properties.

Previous observational studies of the macrophysical properties of cumulus cloud fields have used images from aircraft, space shuttle, and land-imaging satellite instruments. These are summarized in Table 1. Most of these studies are based on small observational data sets, with resolutions of 30–60 m and at most 19 scenes. Only two of these studies (Zhao and Di Girolamo, 2007; Mieslinger et al., 2019) used high-resolution satellite data at 15 m with a larger number of scenes. Moreover, among these previous studies, the loca-

tions analyzed are very sparse. Thus, there remains a need to get more statistics on the macrophysical properties of cumulus cloud fields using high-resolution satellite data in other regions, particularly with well-characterized environments obtained, for example, with intensive field campaigns. Note that the characterization of the aerosol environment is also of importance given its impact on cloud macro- and microphysical properties (e.g., McFarquhar et al., 2004; Yuan et al., 2011; Li et al., 2010; Sheffield et al., 2015).

Caution must be taken when comparing cumulus cloud statistics using different scales, given that different instruments have different domain sizes and spatial resolutions as shown in Table 1. A study by Dey et al. (2008) showed how the statistics of the macrophysical properties of trade wind cumuli derived by Zhao and Di Girolamo (2007) changed with domain size and pixel resolution. As the domain size decreases, the probability of separating clouds into smaller clouds increases and the probability of having cloudier and clearer domains increases. As the pixel resolution degrades, smaller clouds amalgamate into larger clouds and the number of partially filled cloudy pixels increases. Moreover, when comparing aircraft with satellite data, differences between the one-dimensional and two-dimensional measurements of cloud size distributions must be considered. Aircraft cloud size distributions are usually biased towards smaller cloud sizes, assuming random sampling (Rodts et al., 2003; Romps and Vogelmann, 2017; Barron et al., 2020). As a result, differences in time, location, instrument resolution, and sampling must be considered when comparing between different studies.

Table 1 also summarizes the properties of cumulus clouds based on LES models. Low-cloud feedback has been studied by using simulations of clouds from LES models to find a relationship between cloud properties and large-scale meteorology that can be used to predict how clouds will respond to changes in the meteorology within coarser-scale models. Most of these studies have focused on stratocumulus clouds. Some of these findings, for example, indicate that the lower-tropospheric stability (LTS), estimated inversion strength (EIS; Wood and Bretherton, 2006; McCoy et al., 2017), reduced subsidence (Myers and Norris, 2013; Blossey et al., 2013; van der Dussen et al., 2016), sea surface temperature (Qu et al., 2015; Stein et al., 2017; Geiss et al., 2020; McCoy et al., 2017), and surface wind speed (Bretherton et al., 2013) all can have an impact on cloud cover and cloud top height. Fewer studies have been done on shallow cumulus clouds. In particular, Nuijens and Stevens (2012) and Brueck et al. (2015) found that surface wind speed can influence cloud amount and cloud top height. Yamaguchi et al. (2019) have also shown that vertical wind shear can influence cumulus deepening. Given the limited number of studies, there is a need not only for more observations of cumulus clouds but also for their relationships with large-scale meteorological conditions (Klein et al., 2017).

Here, we provide a study of cloud macrophysical properties using data collected during the National Aeronautics and Space Administration (NASA) Cloud, Aerosol and Monsoon Processes Philippines Experiment (CAMP²Ex; Reid et al., 2023). This field campaign offered an opportunity to investigate shallow cumulus clouds in a different region from those studies listed in Table 1. The CAMP²Ex mission took place in the Philippines from August to October 2019, with the goal of investigating cloud–aerosol interactions and their influence on the southwest monsoon precipitation in the region. During CAMP²Ex, the Advanced Spaceborne Thermal Emission and Reflection Radiometer (ASTER) was tasked to sample clouds over parts of the ocean near the Philippines region. ASTER provides high-resolution data suitable for studying cloud macrophysical properties of cumulus cloud fields.

This study uses 170 ASTER scenes dominated with cumulus clouds and collected during CAMP²Ex to analyze the macrophysical properties of cumulus cloud fields and their relationships with larger-scale meteorological conditions in the tropical western Pacific. The ASTER data used in this study are discussed in Sect. 2. Section 3 discusses the cloud masking technique used. Section 4 presents the results of trade wind cumulus properties, which include cloud size, cloud fraction, cloud area–perimeter relationship, cloud top height, and spatial distribution. Cloud fraction and cloud top height are also compared to those derived from the Multiangle Imaging SpectroRadiometer (MISR) and Moderate Resolution Imaging Spectroradiometer (MODIS). Section 5 examines the relationship of the macrophysical properties observed with the meteorological conditions, while Sect. 6 summarizes our results.

## 2 ASTER data

ASTER is an instrument on board the NASA Earth Observing System Terra spacecraft, which is in a Sun-synchronous orbit with an Equator crossing time of around 10:30 local time. ASTER has two cameras: one that points at nadir and one that points backward in the along-track direction. The nadir camera has three visible and one near-infrared spectral bands (0.50 to 1.0 µm) at 15 m spatial resolution, six shortwave infrared spectral bands (1.0 to 2.5 µm) at 30 m spatial resolution, and five thermal infrared spectral bands (8 to 12 µm) at 90 m spatial resolution. Note that the ASTER shortwave infrared data are no longer available after April 2008 due to the detectors not functioning. ASTER takes around 600 scenes in a day, with each scene covering a 60 km × 60 km area. More details about the ASTER instrument can be found in Abrams (2000).

Since ASTER was designed for land surface studies, it primarily collects data over land only. However, as part of the CAMP²Ex mission, it was tasked to acquire data over the Philippines region (0–25° N, 110–135° E) from August to October 2019 with appropriate radiometric gain settings for acquiring cloud properties. The ASTER Level 1T (L1T) calibrated radiance data (version V003) that were collected during the mission were used in this study. In total, there were 2022 scenes from 81 separate days. Scenes occurring over land were not included to facilitate the cloud masking described below, giving a total of 1217 oceanic scenes. There were a lot of cirrus present during the observing period (Reid et al., 2023). Scenes that had any pixel with a brightness temperature less than 0 °C were also excluded to avoid potential cirrus contamination, leaving a total of 378 scenes. These too may contain cirrus, as described in the next section.

## 3 Cloud masking and labeling

To generate cloud masks, a single threshold approach was applied to the ASTER 3N channel (760–860 nm, nadir view) over each 60 km × 60 km scene, following earlier studies (Zhao and Di Girolamo, 2006, 2007; Dey et al., 2008; Jones et al., 2012; Dutta et al., 2020). In brief, a threshold was manually selected for each scene by simultaneously viewing the radiance and mask images to visually verify the chosen threshold. Any scene that was not easily masked with a single threshold was removed from the data set. Some scenes that contained sun glint were used if a threshold could be accurately determined, while scenes that were visually determined to be contaminated by cirrus were not used in the analysis. Thresholds were successfully obtained for 180 ASTER scenes that were visually determined to be dominated by cumulus clouds. Figure 1 shows an example of an ASTER scene on 12 August 2019 along with its cloud mask.

Scenes that contained clouds with cloud area-equivalent diameters, defined as the diameter of a circle with the same cloud area, greater than 30 km, or half the ASTER scene domain, were excluded from the analysis. This was done to reduce the uncertainty in cloud edges with the finite domain size being 60 km × 60 km. Moreover, clouds of this size are typically not classified as trade cumulus. This reduced the total number of scenes to 170 from 36 separate days. There were 46, 33, and 91 scenes for the months of August, September, and October, respectively. Figure 2 shows a map of the CAMP²Ex region, with the centers of each of the 170 ASTER scenes colored by the date each scene was collected. The final list of the ASTER scenes and the thresholds derived for each scene can be found in the Supplement, thereby making our results reproducible.

At the 15 m spatial resolution, each pixel was assigned as either completely cloudy or clear. After classification, pixels were grouped into individual clouds using the four-connectivity rule, where two cloudy pixels that share one edge but not one vertex belong to the same cloud. This was done using the cloud labeling algorithm available in Zhao (2006). In total, the number of clouds found within the 170 scenes is 2 181 059.

**Table 1.** Summary of cumulus macrophysical properties from past studies using observations (o) and models (m). The cumulus macrophysical properties from this study are highlighted in bold text.

| Reference | Instrument | Domain (km × km) | Spatial resolution (m) | Data type | Location/study | No. of scenes | Sub-scenes?[c] | n(x) | λ1 | λ2 | Dc (km) | d1 | d2 | dc (km) | Average cloud fraction |
|---|---|---|---|---|---|---|---|---|---|---|---|---|---|---|---|
| Plank (1969) | camera on aircraft | 16 × 32 | NR | o | Florida coast | 12 | yes | — | — | — | — | — | — | — | 0.25 |
| Wielicki and Welch (1986) | MSS[d] | 170 × 185 | 57 | o | United States, tropical western Atlantic, western Arkansas, Gulf of Mexico | 4 | yes | — | — | — | — | — | — | — | 0.15–0.19 |
| Cahalan and Joseph (1989) | MSS | 170 × 185 | 57 | o | Pacific, South America, Florida coast | 16 | yes | — | — | — | — | 1.27 | 1.55 | 0.5 | 0.45 |
| | TM[e] | 65 × 65 | 28.5 | o | | 19 | yes | logD | 0.89 | 2.76 | 0.5 | 1.34 | 1.47 | 0.5 | 0.55 |
| Sengupta et al. (1990) | MSS | 170 × 185 | 57 | o | tropical Atlantic, Gulf of Mexico, United States, France | 10 | yes | D | 1.39 | 2.35 | 1 | 1.20–1.27 | 1.50–1.73 | 0.5 | |
| Benner and Curry (1998) | MAS[f] | 37 × 37 | 50 | o | tropical western and central Pacific, Maldives, Somali coast, Coral Sea, Caribbean Sea | 17 | yes | D | 1.98 | 3.06 | 0.9 | 1.23 | 1.374 | 0.5 | 0.0925 |
| | space shuttle | 110 × 145 | 30 | o | | 5 | yes | D | 0.94 | 2.91 | 0.6 | 1.1 | 1.34 | 0.5 | 0.09 |
| Gotoh and Fujii (1998) | TM | 65 × 65 | 28.5 | o | Japan | 1 | yes | — | — | — | — | 1.36 | 1.677 | 0.7 | — |
| Zhao and Di Girolamo (2007) | ASTER | 60 × 60 | 15 | o | tropical western Atlantic | 152 | no | D | 2.85[g] | — | — | 1.28 | — | — | 0.086 |
| | | | | | | | | | 1.88[h] | 3.18 | 0.6 | — | — | — | |
| | | | | | | | | | 2.19[i] | — | — | — | — | — | |
| Jiang et al. (2008) | aircraft | – | 50 | o | Houston | 5 | – | D | 2.3 | — | — | — | — | — | — |
| Koren et al. (2008) | ETM+[j] | 170 × 185 | 30 | o | Bahamas, Barbados, Polynesia, Hawaii, southeast of Ascension Island | 5 | no | logA | 1.3 | — | — | — | — | — | 0.1–0.25 |

| Reference | Data description | | | | | | | Size distribution[a] | | | | Fractal dimension[b] | | | Average cloud fraction |
|---|---|---|---|---|---|---|---|---|---|---|---|---|---|---|---|
| | Instrument (km × km) | Domain (km × km) | Spatial resolution (m) | Data type | Location/study | No. of scenes | Sub-scenes?[c] | $n(x)$ | $\lambda_1$ | $\lambda_2$ | $D_c$ (km) | $d_1$ | $d_2$ | $d_c$ (km) | |
| Mieslinger et al. (2019) | ASTER | 60 × 60 | 15 | o | tropical central and eastern Pacific, tropical western Atlantic | 1158 | no | logD | 2.55 | – | – | 1.19 | – | – | 0.087 |
| Luebke et al. (2022) | GOES-16[k] | – | 2000 | o | Barbados | 842 | yes | – | 1.68 | 3.12 | 0.59 | – | – | – | 0.204 |
| **This study** | **ASTER** | **60 × 60** | **15** | **o** | **tropical western Pacific** | **170** | **no** | **D** | **2.93[g]** | **–** | **–** | **1.25** | **–** | **–** | **0.115** |
| | | | | | | | | | **1.95[h]** | **3.27** | **0.6** | **1.20** | **1.55** | **0.5** | |
| | | | | | | | | | **2.16[i]** | **–** | **–** | **–** | **–** | **–** | |
| Neggers et al. (2003) | LES | 6.4 × 6.4 | 50 | m | BOMEX[l] | – | – | D | 1.7 | – | 0.7 | – | – | – | 0.14 |
| | | | 50 | | SCMS[m] | – | – | | – | – | 1.05 | – | – | – | |
| | | | 66.67 | | SGP-ARM[n] | – | – | | – | – | 0.4–1.25 | – | – | – | |
| Heus and Selfert (2013) | LES | 25 × 25 | 25 | m | RICO[o] | – | – | D | – | 2.2 | 0.4 | – | – | – | 0.138 |
| Dawe and Austin (2012) | LES | 6.4 × 6.4 | 25 | m | BOMEX | – | – | D | 1.88 | – | 1 | – | – | – | – |

[a] Power law defined by Eq. (1). [b] Power law defined by Eq. (5) for area perimeters. [c] Manually selected subsets of larger scenes that were cut to show only the cumulus-dominated area. [d] Landsat Multispectral Scanner (MSS). [e] Landsat Thematic Mapper (TM). [f] Moderate Resolution Imaging Spectrometer (MODIS) Airborne Simulator (MAS). [g] Single-line least-squares fit. [h] Double-line least-squares fit with a scale break. [i] Direct power law fit. [j] Enhanced Thematic Mapper Plus (ETM+). [k] Geostationary Operational Environmental Satellite-16 (GOES-16). [l] Barbados Oceanographic and Meteorological EXperiment (BOMEX). [m] Small Cumulus and Microphysics Study (SCMS). [n] Southern Great Plains (SGP) site of the Atmospheric Radiative Measurement (ARM). [o] Rain in Cumulus over the Ocean (RICO). NR means not reported.

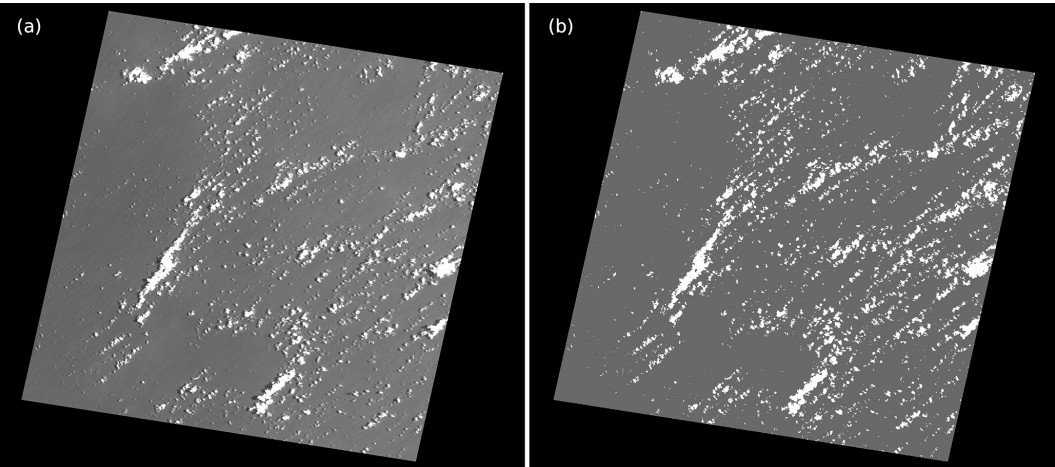

**Figure 1. (a)** ASTER channel 3N image (60 km × 60 km) taken on 12 August 2019 and **(b)** its cloud mask, where white represents cloud, grey represents clear, and black represents no data.

The uncertainties in our cloud masking approach and their impact on the cumulus statistics are discussed in Zhao and Di Girolamo (2007). The largest impact is on cloud fraction, where much of the uncertainty from our approach comes from cloud edge pixels. This uncertainty can be estimated using the formulation given by Di Girolamo and Davies (1997). Otherwise, cumulus cloud size distributions and spatial distributions have been shown to be insensitive to small perturbations in the choice of cloud detection thresholds as shown and discussed in other similar studies (e.g., Wielicki and Welch, 1986; Zhao and Di Girolamo, 2007).

## 4   Observed statistics of cumulus cloud macrophysical properties

The following sections present the trade wind cumulus macrophysical properties over the tropical western Pacific using the 170 cloud-masked scenes from ASTER channel 3N that were derived in Sect. 3. These include cloud size, cloud fraction, cloud area–perimeter relationship, cloud top height, and spatial distribution. These properties are compared to previous studies on oceanic shallow cumulus cloud properties and, in the case of cloud fraction and cloud top height, to MISR and MODIS satellite retrievals.

### 4.1   Cloud size distribution

The cloud size distribution shows the fraction of clouds within a finite range of sizes. The cloud size distribution has been commonly observed to follow a power law distribution (Benner and Curry, 1998) and has been used to compare cloud models and observations (Neggers et al., 2003). The cloud size distribution $n(D)$ following the power law is given by

$$n(D) = aD^{-\lambda}, \tag{1}$$

where $D$ is the cloud area-equivalent diameter, and $a$ and $\lambda$ are constants. The area of each cloud is defined as the product of the number of cloudy pixels and the area of each pixel. $D$ is then calculated from the cloud area by assuming a perfectly circular cloud.

The scaling parameter, $\lambda$, can be determined by taking the natural logarithm of both sides of Eq. (1), giving

$$\ln n(D) = \ln(a) - \lambda \ln D, \tag{2}$$

where $\lambda$ is the slope of the least-squares linear regression between $\ln n(D)$ and $\ln D$. This method of fitting a least-squares line, called the "line fit" method, has been shown to give more weight to larger clouds, which tend to be poorly sampled, and is sensitive to the binning strategy. Thus, the "direct power law fit" method described by Zhao and Di Girolamo (2007) was also used in this study.

In the direct power law fit method, the mean of all cloud diameters $\overline{D}$ is first determined as

$$\overline{D} = \frac{1}{n} \sum_i^n D_i, \tag{3}$$

where $n$ is the total number of clouds in a scene, and $D_i$ is the area-equivalent diameter of each cloud. From Eq. (1), the probability density function of $D$ is given as $f(D) = (\lambda - 1)D^{-\lambda}$, so that the expected value of $D$, $E(D)$, is

$$E(D) = \frac{\int_{D_0}^{D_u} D f(D) \, dD}{\int_{D_0}^{D_u} D \, dD} = \frac{\int_{D_0}^{D_u} D^{1-\lambda} \, dD}{\int_{D_0}^{D_u} D^{-\lambda} \, dD}$$
$$= \frac{(1-\lambda)(D_u^{2-\lambda} - D_0^{2-\lambda})}{(2-\lambda)(D_u^{1-\lambda} - D_0^{1-\lambda})}, \tag{4}$$

TS1 where $D_0$ and $D_u$ are the smallest and largest cloud diameters among all the clouds, respectively. For a sufficiently

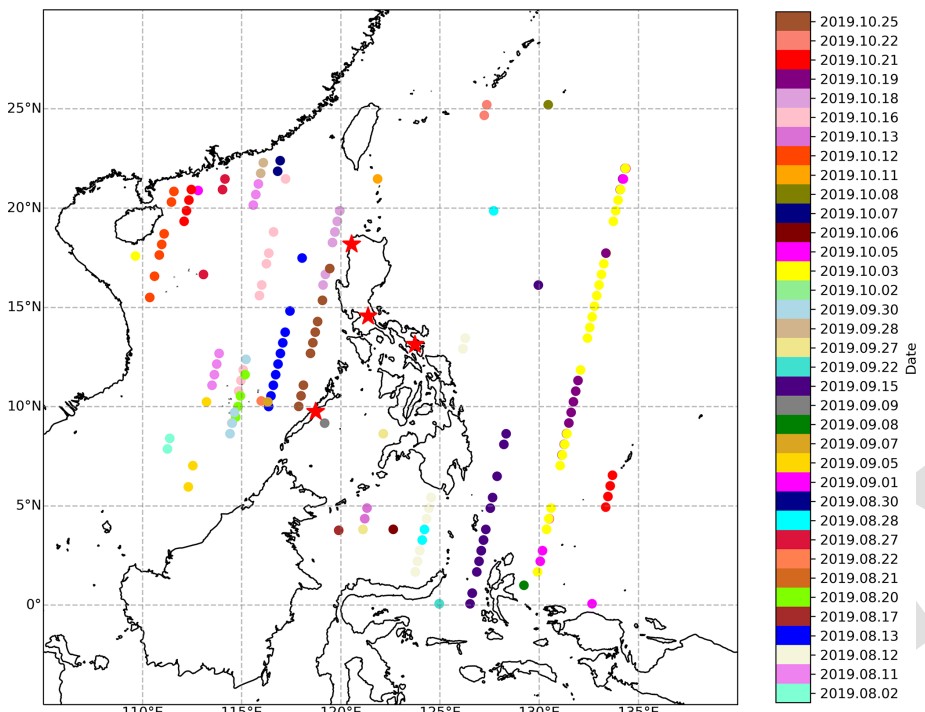

**Figure 2.** The center of each ASTER 60 km × 60 km scene used in the CAMP²Ex region, colored by date (YYYY.MM.DD). Red stars indicate operational sounding stations published by the University of Wyoming (https://weather.uwyo.edu/upperair/sounding.html, last access: 16 March 2022).

large number of samples, $\overline{D} \cong E(D)$, and λ can be solved by combining Eqs. (3) and (4). This method has been shown to be statistically unbiased with equal weight assigned to each data point (Zhao, 2006). Note that the direct power law fit method does not make use of any binned data and is thus independent of binning strategy.

The solid line in Fig. 3 shows the normalized frequency of all clouds with area-equivalent diameters of less than 7 km in 100 m bin widths on a logarithmic scale. Clouds with $D > 7$ km are poorly sampled and were excluded in the analysis of cloud size distribution. We note that 68 of the 170 ASTER scenes used contained clouds with $D > 7$ km. However, clouds with $D > 7$ km only make up less than 0.007 % of all clouds observed.

Figure 3 shows λ = 2.93 with correlation coefficient $R = 0.99$ using the line fit method, while the double power law line fit gives $λ_1 = 1.95$ and $λ_2 = 3.27$, with a scale break at $D_c = 0.6$ km that was computed as the point that leads to the least residual. These results are similar to those reported by Benner and Curry (1998) for the double power law from MODIS Airborne Simulator (MAS) images ($λ_1 = 1.98$ and $λ_2 = 3.06$) and Zhao and Di Girolamo (2007) for the single ($λ = 2.85$) and double power law ($λ_1 = 1.88$ and $λ_2 = 3.18$) as seen in Table 1. Mieslinger et al. (2019) report a slightly smaller exponent with the single power law of λ = 2.55 and double power law of $λ_1 = 1.68$ and $λ_2 = 3.12$ but with a similar scale break at $D_c = 0.59$ km. Note again these results are

insensitive to small changes in the cloud masking threshold used but can differ with other studies due to domain size and spatial resolution.

Figure 3 also shows that the direct power law fit gives λ = 2.16 with $R = 0.99$, which is similar to Zhao and Di Girolamo (2007). This is shown as a dashed step line in Fig. 3 instead of a straight line since the direct power law fit was not calculated based on binned data. A double direct power law fit gives $λ_1 = 2.12$ and $λ_2 = 2.94$, with a scale break at $D_c = 0.4$ km, which is slightly smaller than the scale break computed from the double power law line fit method. Note that the difference between the direct power law fit and the observations for larger cloud diameters ($> 1$ km) is small, on the order of $10^{-4}$, but fits well with the smallest cloud diameters. On the other hand, the difference between the line fit and the observations for the smallest cloud diameters is large, on the order of $10^{-1}$, but fits well for the larger cloud diameters.

## 4.2 Cloud fraction distribution

Figure 4 gives the cloud fraction and cumulative cloud fraction as a function of cloud area-equivalent diameter using bin intervals of 100 m. Cloud fraction was defined as the ratio of the number of cloudy pixels to the total number of pixels. The average cloud fraction from all 170 scenes is $0.115 \pm 0.014$. This uncertainty comes from half the frac-

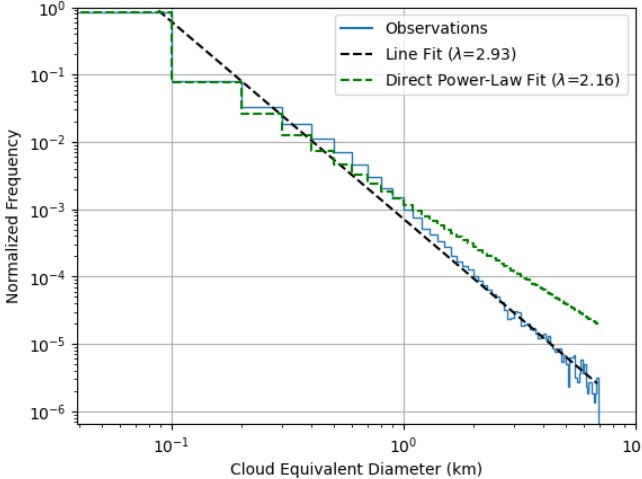

**Figure 3.** Normalized distribution of cloud equivalent diameter of clouds smaller than 7 km in diameter using bin width of 100 m for the 170 ASTER scenes.

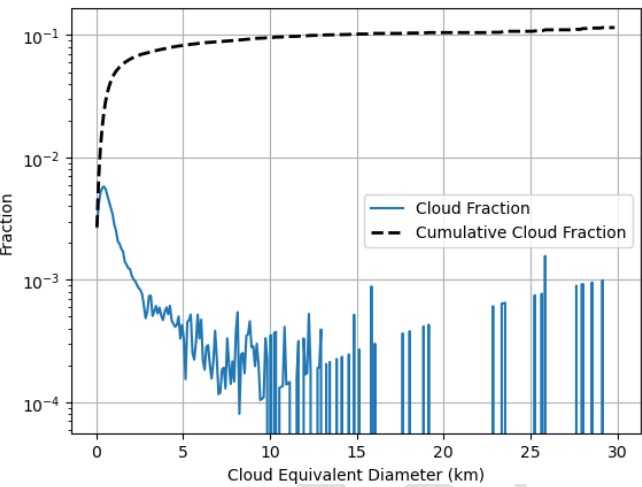

**Figure 4.** Cloud fraction and cumulative cloud fraction as a function of cloud equivalent diameter using bin width of 100 m for 170 ASTER scenes sampled over the western Pacific Ocean during CAMP²Ex.

tion of cloud edge pixels (Di Girolamo and Davies, 1997), which is 0.027. The cloud fraction over the CAMP²Ex region is only slightly larger (around a 0.03 difference) than those from previous studies done over other tropical regions, such as those by Benner and Curry (1998), McFarquhar et al. (2004), Zhao and Di Girolamo (2007), and Mieslinger et al. (2019). Very similar to Zhao and Di Girolamo (2007), half of this total cloud fraction is from clouds less than 1.6 km in diameter. A peak in cloud fraction is observed at cloud diameters between 400 and 500 m, showing how small most of the cumulus clouds sampled in this region are. Note that most of the bins between 10 to 30 km only contained one cloud, so an increase is seen in cloud fraction as cloud area is proportional to cloud diameter.

The ASTER, MISR, and MODIS cloud fractions for the CAMP²Ex region provide an opportunity to compare with results of Zhao and Di Girolamo (2006) and Dutta et al. (2020) in other regions. They showed large overestimates of cloud fraction reported by MISR and MODIS as compared to ASTER. This was due to the presence of subpixel clouds, especially in trade wind cumulus-dominated regions where typical cumulus size can be smaller than the resolution of passive sensors like MISR and MODIS. Here, cloud fractions from the MISR Level 2 Top-of-Atmosphere (TOA)/Cloud Classifiers product (TC_Classifiers) version 3 (Moroney et al., 2014) and from the MODIS Cloud Mask (MOD35) Collection 6.1 (Ackerman and Frey, 2015) were compared to the total cloud fraction from the ASTER scenes. MISR provides two different cloud fraction products, the standard estimate cloud fraction (SECF) and resolution-corrected cloud fraction (RCCF), at 17.6 km resolution, both of which are derived from the Radiometric Camera-by-camera Cloud Mask over ocean (Zhao and Di Girolamo, 2004). To compare ASTER, MISR, and MODIS cloud fractions, ASTER (15 m)

and MODIS (1 km) cloud mask pixels were collocated to the MISR 17.6 km cloud fraction pixels. Cloud fractions from ASTER and MODIS were then calculated for each 17.6 km × 17.6 km region to directly compare them to MISR SECF and RCCF. Note that the MODIS pixels are labeled as cloudy, probably cloudy, probably clear, clear, or no retrieval. Cloudy and probably cloudy pixels were considered as cloudy, while clear and probably clear pixels were considered as clear in computing for the MODIS cloud fraction. This is consistent with how cloud fraction climatologies are derived from MODIS (Stubenrauch et al., 2012). Only regions that contained at least 99 % of the total possible number of ASTER pixels in a 17.6 km × 17.6 km region were used for the comparison. Taking an average over all 17.6 km × 17.6 km regions, the mean ASTER cloud fraction was 0.12, the mean MISR SECF was 0.49, the mean MISR RCCF was 0.14, and the mean MODIS cloud fraction was 0.19. Thus, there is an overestimate of cloud fraction due to pixel resolution for MISR SECF and MODIS that is similar to what was reported by Zhao and Di Girolamo (2006) for scenes over the tropical western Atlantic. We see that the resolution correction provided by the MISR RCCF algorithm (Jones et al., 2012) carries a small bias (+0.02), which is well within the uncertainties reported in this product by Dutta et al. (2020) and very close to the fraction of cloud edge pixels reported above.

## 4.3 Cloud area–perimeter relationship

The scaling relationship between cloud perimeter and cloud area was given by Lovejoy (1982) as

$$P \propto \sqrt{A^d}, \tag{5}$$

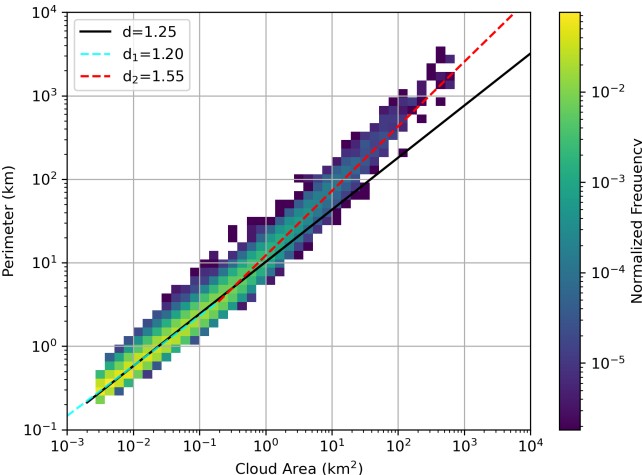

**Figure 5.** Density plot of cloud area against cloud perimeter using 50 logarithmic bins from 0.1 to 10 000 km on the $y$ axis and 0.001 to 10 000 km$^2$ on the $x$ axis for clouds larger than 12 pixels in the 170 ASTER scenes during the CAMP$^2$Ex period with fractal dimension $d$ without a scale break and $d_1$ and $d_2$ with a scale break.

where $A$ is the cloud area, $P$ is the cloud perimeter, and $d$ is the fractal dimension. The value of $d$ describes the complexity of the cloud shape. For regular shapes, such as circles and squares, $d$ is unity, while $d$ approaches the value of 2 as the perimeter becomes more contorted. The perimeter of each cloud was defined as the total length of all the edges adjacent to noncloudy pixels, while the cloud area was computed in the same way as described in Sect. 4.1.

Figure 5 shows the observed normalized frequency of cloud perimeter versus cloud area in log–log space. Using a least-squares fit, the slope of the line was $d = 1.25$ with a correlation coefficient of 0.98. This is slightly smaller than previous studies listed in Table 1, indicating cumulus clouds of smoother shapes were sampled during the CAMP$^2$Ex period. When two lines of best fit were determined, the slopes were $d_1 = 1.20$ and $d_2 = 1.55$, with a scale break in cloud area of $d_c = 0.19$ km$^2$ (or 0.50 km in area-equivalent diameter), which has the least residual. This shows that the larger clouds have more complex shapes than smaller clouds, consistent with previous studies (Cahalan and Joseph, 1989; Benner and Curry, 1998; Gotoh and Fujii, 1998). Note that the fractal dimension $d$ will also be insensitive to small perturbations in the cloud threshold as Cahalan (1991) showed only a 0.1 increase in dimension with around a 300 % increase in threshold.

## 4.4 Spatial distribution

We characterize the spatial distribution using the statistics of the nearest-neighbor distance (NND) in the observed cloud fields as done in Zhao and Di Girolamo (2007). The distance between two clouds was calculated as the Euclidean distance between their mass centers. Figure 6a shows the observed frequency distribution of NND with a peak around 0 to 50 m. However, because the size of a cloud limits the minimum possible value of its NND, Fig. 6b shows the frequency distribution of the ratio of NND to the cloud area-equivalent radius. This shows that around 30 % of clouds have a nearest neighbor within a distance of 3 times their radius. At a bin width of 10 (not shown here), more than 75 % of clouds have a nearest neighbor within a distance of 10 times their radius. This result is very similar to that reported by Zhao and Di Girolamo (2007).

## 4.5 Cloud top height distribution

ASTER channel 14 (12 μm) has the least amount of water vapor absorption among the thermal infrared channels of ASTER. Thus, channel 14 data were used to retrieve the cloud top height (CTH) for each 90 m resolution cloudy pixel using the same method as Zhao and Di Girolamo (2007). A 90 m resolution cloud mask was first constructed by flagging a 90 m resolution pixel in the channel 14 scene as cloudy only if all the 15 m subpixels within the corresponding channel 3N scene were cloudy based on the 15 m cloud mask (Sect. 2.4). For these 90 m cloudy pixels, the brightness temperature (BT) was calculated from the radiance for each cloudy pixel using the procedure in ASTER's Algorithm Theoretical Basis Document for Brightness Temperature Version 3.0 (Alley and Jentoft-Nilsen, 1999). The BT is converted to a CTH by equating it with the temperature profile from a sounding. Note that this approach assumes that cloud emissivity is equal to 1 and that atmospheric absorption of radiation within this channel does not happen above the cloud top.

The soundings used were obtained during the CAMP$^2$Ex mission. These soundings included dropsondes from the NASA P-3 aircraft, as well as ship sondes (van Diedenhoven et al., 2022). However, they were limited in that there were only 11 out of 36 ASTER scene days when there was at least one sounding available. Thus, standard synoptic soundings acquired over the Philippines from four sites, namely stations 98223, 98433, 98444, and 98618 (shown as red stars in Fig. 1) for 00:00 UTC, were also used. As these soundings had lower vertical resolution compared to those from the CAMP$^2$Ex mission, the temperature profile was interpolated to every 5 m. The sounding collected nearest in time and space to that of the ASTER center was used to convert the BT into CTH by equating it with the temperature profile of the sounding. Note that small perturbations to the choice of the cloud detection threshold will only impact the number of 90 m pixels used to retrieve CTH but not the values of CTH (Zhao and Di Girolamo, 2007).

As an estimate of the uncertainty in retrieving CTH due to sounding choice, temperature profiles from each sounding from the aircraft dropsonde was compared to the temperature profile from an operational sounding nearest in time and space. Both temperature profiles were then re-gridded onto an evenly spaced grid from 500 to 4000 m – the altitude

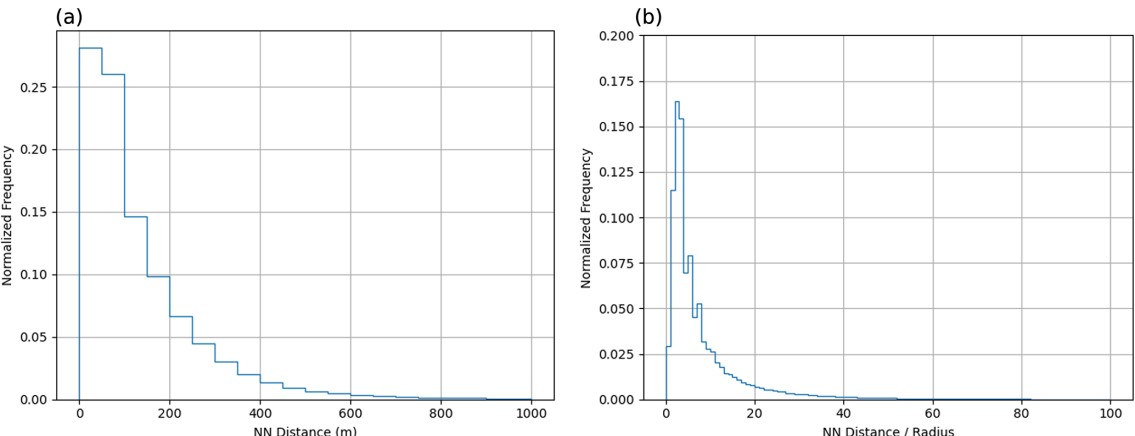

**Figure 6. (a)** Normalized distribution of the nearest-neighbor distance using bin width of 50 m and **(b)** of the ratio of the nearest-neighbor distance to the cloud area-equivalent diameter using bin width of 1 for the 170 CAMP$^2$Ex ASTER scenes.

range containing most of the cloud tops in our data. From this, the root mean square difference (RMSD) between the temperature profiles from the dropsonde and the operational sounding was calculated. An overall mean RMSD of 0.9 K
was calculated. The same was done for each sounding from the ship sondes, leading to an overall mean RMSD of around 1.0 K. With a ±1.0 K uncertainty in the temperature profile due to sounding choice, there is around a ±160 m uncertainty in height. Note that the dropsonde has an uncertainty of
±0.2 K (UCAR, 2020) in temperature measurements, while the ship sonde has an uncertainty of ±0.3 K (Vaisala, 2020). The overall uncertainty in height is then mostly due to sounding choice.

We also want to compare the CTH obtained from the
15 ASTER scenes to that retrieved by other satellite instruments, such as MISR and MODIS. MISR CTH data in the MISR Level 2 TOA/Cloud Height and Motion Parameters product, namely TC_Cloud (Version F01_0001; Mueller et al., 2013), and MODIS CTH data in the MODIS Level 2
Cloud product, namely MOD06 (Version 6.1; Platnick et al., 2015), were retrieved for each ASTER scene by taking the valid MISR CTH retrievals at 1.1 km resolution and the valid MODIS CTH retrievals at 1 km resolution within the ASTER scene. CTH distributions for MISR and MODIS within the
ASTER scenes were then compared to those obtained from the ASTER scenes using the BT technique.

Figure 7a shows the CTH frequency distribution with bin width of 100 m for different ranges of cloud area-equivalent diameter normalized by the total number of the cloudy pixels
examined. The cloud diameters used in Fig. 7a were calculated from the 90 m cloud mask in the same way as calculated from the 15 m cloud masks. On average, cloudy pixels from larger clouds have higher cloud tops than those from small clouds as seen in the widening of curves as the range of cloud
diameters increase. A peak in ASTER-retrieved CTH around 1.2 to 1.3 km was observed from all clouds.

As a comparison, Fig. 7b shows the CTH distribution with bin width of 250 m for all ASTER scenes versus the MISR and MODIS CTHs retrieved for each scene. Figure 7b also shows the CTH distribution derived from the High-Spectral-
40 Resolution Lidar – Generation 2 (HSRL-2; Hair et al., 2008; Burton et al., 2018) instrument on board the NASA P-3 aircraft during CAMP$^2$Ex (Fu et al., 2022), which were not typically coincident with the ASTER scenes. The MISR CTH distribution peaks around 750 to 1000 m, the MODIS CTH
distribution peaks around 0 to 250 m, and the HSRL-2 CTH distribution peaks around 500 to 750 m. Thus, there is around a 1250 m spread in the location of the peak in the CTH distributions between the four instruments. Below we discuss this further to better understand the likely causes for the differ-
ences.

First, the CTH distribution from in situ measurements using the HSRL-2 (data processed by Fu et al., 2022) is seen to have two modes, having CTHs larger than 4 km, as compared to the CTH distribution from ASTER. Recall that ASTER
(and coincident MISR and MODIS) scenes do not include scenes that have any clouds colder than 0 °C anywhere in a 60 km × 60 km ASTER scene. No such filtering was placed on the HSRL-2 data, which include all heights observed. The 0 °C isotherm is around 5 km, yet heights greater than 4 km
are still observed from the HSRL-2 data, albeit very infrequently. This may be due to a sampling bias towards larger clouds in the aircraft data, as larger clouds, such as congestus, were targeted during research flights (Fu et al., 2022). Still, it is the lower clouds that dominate the frequency distribution.
Since the HSRL-2 data are a sample from 19 separate days scattered over the ASTER collection period, the CTH mode in the 500–750 m bin is taken as reference for comparison. We see that this lower peak is closer to the peak in MISR CTHs than it is to the peak in ASTER CTHs.
The MODIS CTH distribution peak is around 1000 m lower than the peak in the BT-derived ASTER CTHs, 750 m

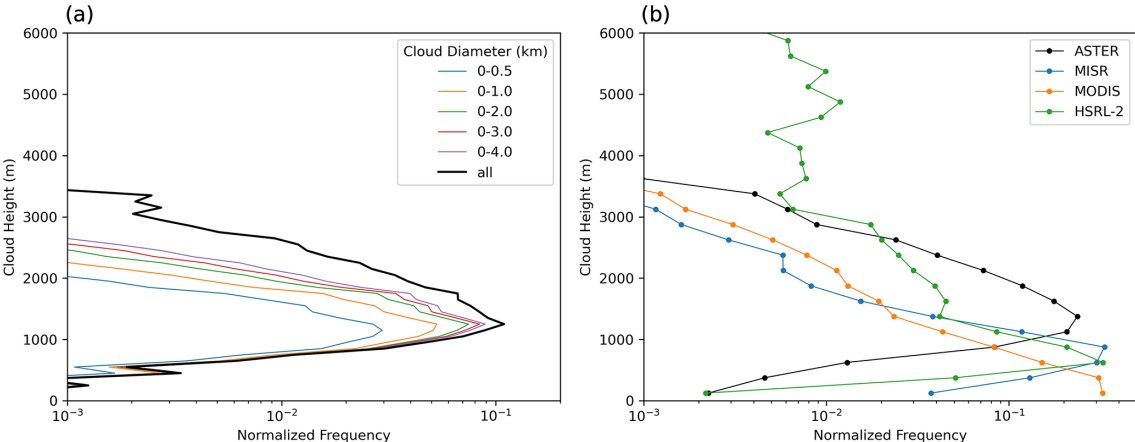

**Figure 7. (a)** Normalized distribution of cloud top heights with 100 m bin width from ASTER for different cloud equivalent diameters and **(b)** a comparison of the normalized distribution of cloud top heights with 250 m bin width obtained from ASTER, MISR, and MODIS corresponding to each ASTER scene and from the HSRL-2 on board the P-3 aircraft.

lower than the peak in the MISR stereoscopically retrieved CTHs, and 500 m lower than the peak in the HSRL-2. When collocated ASTER and MODIS pixels were examined, 81 % of the 1 km MODIS pixels reporting CTH had CF < 0.50 based on the ASTER 15 m mask (See Fig. 9 of Reid et al., 2023, for an overlay of an ASTER–MODIS example highlighting this effect). This is consistent with the results of Sect. 4.2, where a peak in the cloud fraction distribution was seen at the 400 to 500 m cloud equivalent diameter bin. Because of its coarse resolution, the CTH MODIS retrieval is affected more by the surface temperature when there are subpixel clouds compared to ASTER, thus biasing the cloud height low.

MODIS also uses a latitude-dependent BT lapse rate to infer low-level cloud heights over water. These lapse rates were developed from collocated MODIS-observed 11 μm BTs, CALIPSO low-level cloud heights, and sea surface temperatures, with a different set of lapse rates determined for each month, to create the monthly zonal mean "apparent 11 μm BT lapse rates" (Baum et al., 2012). Thus, differences in the lapse rate used by MODIS and the "true" lapse rate from the sounding can also cause differences in the retrieved CTH distributions between MODIS and ASTER. Using the coefficients given in Baum et al. (2012), the lapse rate used by MODIS for each ASTER scene was computed and averaged to $3.34\,\mathrm{K\,km^{-1}}$ for all scenes. Lapse rates were also computed from the surface to around 2 km in altitude for the sounding used for each ASTER scene in retrieving CTHs. This averaged to $5.74\,\mathrm{K\,km^{-1}}$ for all scenes. Thus, there is a $2.40\,\mathrm{K\,km^{-1}}$ difference in lapse rates, with MODIS having a lower lapse rate. This would then translate to MODIS reporting CTHs with a high bias of around 800 m on average, which is opposite to what is observed in Fig. 7b. This strongly implies the dominating effect of subpixel clouds

is leading to an overall lower MODIS CTH relative to the ASTER, MISR, and HSRL-2.

For the MISR CTH distribution, the peak frequency occurs in the 750–1000 m bin at 0.33, although the frequency in the 500–750 m is only slightly lower at 0.30 as seen in Fig. 7b. This lines up well with the HSRL-2, having a peak CTH frequency in the 500–750 m bin but with a frequency in the 750–1000 m bin at 0.21, which is not much different from the MISR peak at 0.33. The slight difference may be an indication of sampling differences between MISR and HSRL-2 observations. Note that MISR retrieves CTH using a stereoscopic technique (Mueller et al., 2013), which is not impacted by subpixel cloud biases. As such, the difference between the ASTER and MISR CTHs is likely due to violations in the assumptions of the ASTER BT retrieval technique. One of the assumptions is that the cloud emissivity is always equal to 1, which may not always be true. While CTH retrievals were from channel 14 90 m pixels that were fully cloudy based on the 15 m cloud mask, some 15 m cloudy pixels may not have been fully cloud covered. Since the ocean surface is warmer than clouds in this region and time period, optically thin cloud or subpixel cloud would bias the cloud top heights low, opposite to what is observed relative to the MISR or HSRL-2 CTHs. This leads to the possibility of either significant thin cirrus contamination or significant water vapor absorption as the possible source for ASTER CTHs that are biased high relative to MISR and HSRL-2. Thin cirrus contamination was ruled out because of the care taken in filtering scenes that contain cirrus, including the use of the 12 μm channel. To examine the impact of water vapor absorption, soundings used for each scene, along with the spectral response function of ASTER channel 14 (ASTER instrument characteristics, https://asterweb.jpl.nasa.gov/characteristics.asp, last access: 2 July 2023), were input into the radiative transfer model li-

bRadtran (Mayer et al., 2020) with a cloud layer of optical depth 9.6 at 700 to 900 m in altitude to simulate BTs retrieved. Simulations without water vapor in the atmosphere were run to examine the impact of water vapor absorption.

Results showed that a 2 to 5 K cooling due to water vapor absorption can lower the ASTER-retrieved CTHs by $\sim 200$ to 900 m, thus bringing it more in line with the MISR heights, which had a mode that was 500 m lower than ASTER.

In summary, Fig. 7b shows the different CTH distributions retrieved using different techniques. The HSRL-2 CTH distribution has a peak close to that of MISR, with the small differences between the two likely due to sampling differences. The MODIS CTH peak is the lowest among the distributions, where the presence of subpixel clouds leads to the overall lower MODIS CTHs. The ASTER CTHs shown in both Fig. 7a and b do not include any correction for water vapor absorption; thus they are biased high. Radiative transfer simulations indicate that bias expected from the neglect of water vapor absorption in this region and time period is consistent with the difference we see between ASTER and MISR CTHs. Finally, the analyses above were for coincident MISR and MODIS data that fell within the ASTER 60 km × 60 km scenes. Not all ASTER clouds had retrieved MISR and MODIS CTHs. When restricting the analysis (not shown) to pixels with valid MISR and MODIS CTHs that had at least one ASTER CTH, none of the conclusions drawn above changed.

The CTHs reported here are also similar to the heights reported for the RICO region over the Atlantic east of Antigua (Zhao and Di Girolamo, 2007). They too used ASTER for retrieving CTH and found a peak frequency around 900 m. They reported that neglecting water vapor absorption could bias the cloud heights up to 200 m. This suggests that total column water vapor (TCWV) observed during CAMP[2]Ex was much higher than that observed during RICO. Indeed, Fig. 8 shows the TCWV for the CAMP[2]Ex and RICO regions during their respective mission periods, as derived from the fifth generation of atmospheric reanalysis data (ERA5) by the European Centre for Medium-Range Weather Forecasts (ECMWF; Hersbach et al., 2020). Repeating the radiative transfer simulations above, results showed a $\sim 2$ K BT difference between the two regions. Thus, accounting for water vapor absorption in estimating CTH from ASTER, both regions show a peak frequency in CTH of $\sim 750$ m.

## 5   Comparison of meteorological variables with macrophysical properties

In the previous sections, the macrophysical properties (size, areal fraction, area–perimeter relationship, CTH, and a metric for the spatial distribution) of boundary layer cumuli observed during CAMP[2]Ex using ASTER were presented and compared to in situ measurements, prior studies, and retrievals from other satellites. In this section, the relationship

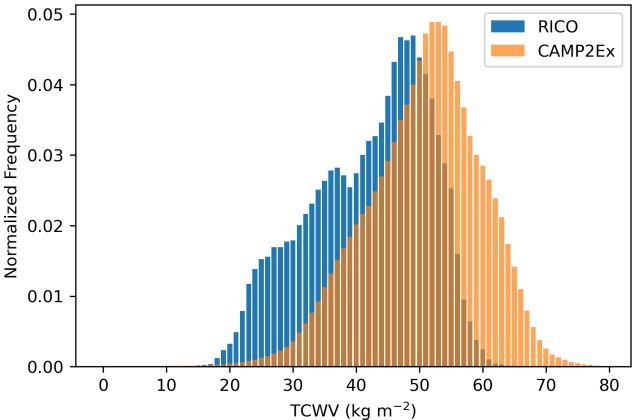

**Figure 8.** Normalized distribution of total column water vapor from ERA5 reanalysis data for the RICO (blue) and CAMP[2]Ex (orange) regions.

of those cumulus cloud macrophysical properties to large-scale meteorological parameters is examined.

Previous studies have investigated the relationship between the large-scale meteorology and the shallow cumulus cloud macrophysical properties. Zhao and Di Girolamo (2007) did not find any relationship between the two during the RICO campaign, which they hypothesized was due to the inability of the forecast meteorological data that they used to capture the scales in which the cumulus clouds reside. Mieslinger et al. (2019), on the other hand, analyzed cumulus cloud fields over the western Atlantic and the central and eastern Pacific and found that the surface wind speed was the dominant controlling factor for their estimated cloud macrophysical properties. They found that with increasing surface wind speed, the cloud fraction and cloud top heights increased, leading to a shift in the cloud size distribution toward larger clouds with smoother shapes. They reached this conclusion by examining the average cloud field properties as a function of single-binned meteorological parameters using ERA5 Interim data. However, data binning can create artificial relationships between uncorrelated data depending on how the data are binned (Wainer et al., 2006; Rusakov, 2023). To avoid this issue, we use multiple linear regression to investigate the relationship between the large-scale meteorology and observed macrophysical properties.

The ERA5 reanalysis data were used to characterize the large-scale meteorological conditions instead of the available soundings due to the limited temporal and spatial coverage of the soundings and inconsistencies in the spatial and temporal resolution between the different sounding instruments. The ERA5 reanalysis data at the hours of 01:00, 02:00, 03:00, and 04:00 UTC were used, the time period that the ASTER scenes were taken. Data included the $u$ (east) and $v$ (north) wind components at 10 m and pressure levels of 1000–850 hPa (every 25 hPa) and 600 hPa (the boundary layer top), vertical velocity and relative humidity at

**Table 2.** Meteorological variables at the different levels used in this study. Fields extracted or calculated from the ERA5 reanalysis data.

| Meteorological variable (units) | Levels |
| --- | --- |
| $u$ and $v$ wind components [m s$^{-1}$] | 1000–850 hPa (every 25 hPa), 600 hPa |
| Vertical velocity [Pa s$^{-1}$] | 1000–850 hPa (every 25 hPa), 600 hPa |
| Relative humidity | 1000–850 hPa (every 25 hPa) |
| Sea surface temperature (SST) [K] | – |
| Total column water vapor (TCWV) [kg m$^{-2}$] | – |
| Wind speed [m s$^{-1}$] | 10 m, 1000–850 hPa (every 25 hPa), 600 hPa |
| Wind speed difference [m s$^{-1}$] | 1000–975 (950, 925, 900, 875, 850) hPa, 950–925 (900, 875, 850) hPa, 850–600 hPa |
| Wind direction [°] | 10 m, 1000–850 hPa (every 25 hPa), 600 hPa |
| Lower-tropospheric stability (LTS) [K] | 1000–975 (950, 925, 900, 875, 850, 700) hPa |
| Estimated inversion strength (EIS) [K] | 1000–700 hPa |
| Equivalent potential temperature [K] | 1000–850 hPa (every 25 hPa) |
| Equivalent potential temperature difference [K] | 1000–975 (950, 925, 900, 875, 850) hPa, 950–925 (900, 875, 850) hPa |
| Convective available potential energy (CAPE) [J kg$^{-1}$] | 850 hPa |
| Convective inhibition (CIN) [J kg$^{-1}$] | 850 hPa |

pressure levels of 1000–850 hPa (every 25 hPa), sea surface temperature (SST), and total column water vapor (TCWV). These variables were compiled for the time period of August to October 2019 over the CAMP$^2$Ex region with a horizontal resolution of $0.25° \times 0.25°$. From these, other variables such as wind speed, wind speed difference, wind direction, lower-tropospheric stability (LTS), estimated inversion strength (EIS), equivalent potential temperature ($\theta_e$), equivalent potential temperature difference, convective available potential energy (CAPE), and convective inhibition (CIN) were calculated using MetPy version 1.4.1 in Python. This gave a total of 88 meteorological variables, which are shown in Table 2. Note that the ERA5 boundary layer height (BLH) variable was also examined and was found to be insignificant based on $p$ value for all the observed macrophysical properties. Therefore, it was not included in the final list of variables shown in Table 2. The low ranking may be due to the decoupling of the planetary boundary layer (PBL) in the trade cumulus regime, leading to relatively shallow and small cumuli with CTHs well below the PBL top as found in other studies (Karlsson et al., 2010; Kubar et al., 2020).

Figure 9 shows contour normalized frequency by altitude diagrams for relative humidity and temperature from the ERA5 reanalysis data. A wide spread in relative humidity is seen, while the temperature has a relatively narrow spread. The median vertical profile is shown as the solid black line. At the surface, there is a median relative humidity value of around 80 % and a temperature of around 300 K. These have around a 9 % and 2 K RMSD relative to the median relative humidity and temperature profile soundings, respectively, obtained from the dropsondes and ship sondes that were used in Sect. 4.5 for CTH retrieval. Comparing the frequency by altitude diagram for relative humidity to a similar figure in Davison et al. (2013) for the RICO campaign, lower relative humidity was generally observed for the RICO field campaign at altitudes above 3 km. This is consistent with the

lower TCWV values observed during RICO from the ERA5 reanalysis data as discussed in Sect. 4.5.

For each ASTER scene, the mean value of the meteorological variables (Table 2) contained within each scene was taken to represent the whole scene. The pooled standard deviation of the meteorological variables contained within each ASTER scene can be found in the Supplement. Standard deviations are relatively small, as seen in the median coefficient of variation values (around 0.1), showing the representativeness of taking the mean to represent the whole scene. The general statistics for the mean meteorological variables among all scenes are also given in the Supplement. In general, there is a wide variability seen in the mean meteorological variables, apart from SST. All 88 meteorological variables were then standardized by subtracting the mean and dividing by the standard deviation.

The different statistics of cloud macrophysical properties were obtained from individual ASTER scenes, and the standardized meteorological variables were then fit into a multiple linear regression model for each cloud macrophysical property. The $R^2$ and adjusted $R^2$ values obtained when doing so are shown in Table 3. The adjusted $R^2$ value considers the number of variables in the model, and unlike the $R^2$ value, it only increases when the new variable improves the model more than would be expected by chance. As seen in the adjusted $R^2$ values, less than 30 % of the variation in the cloud size distribution parameters (line fit $\lambda$ and direct power law fit $\lambda$) and fractal dimension is explained by all the variables. This may be due to the low variability seen in these properties among all the scenes (see the Supplement). However, for the cloud fraction and mean cloud top height, 51 % and 72 % of their variation can be explained by the variables, respectively. With this, the discussion below only focuses on the results for cloud fraction and mean cloud top height.

To rank the meteorological variables that affected the cloud properties, variable selection was first done from the

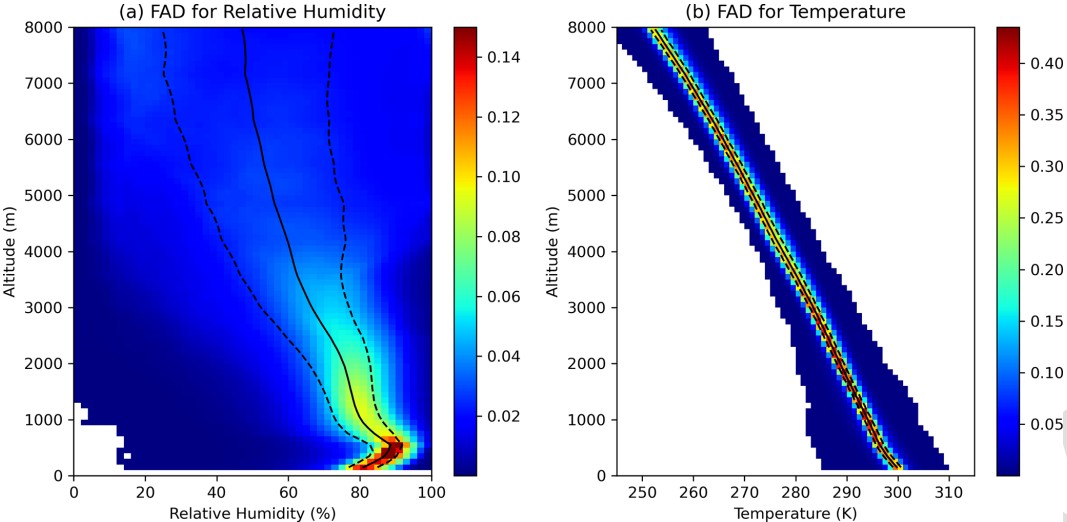

**Figure 9.** Contour frequency by altitude diagrams for relative humidity **(a)** and temperature **(b)** from the ERA5 reanalysis data. Relative humidity bins are 2 %, temperature bins are 1 K, and altitude bins are 100 m. The median is shown by the solid black line, while the 25th percentile and 75th percentile are shown by dotted black lines.

**Table 3.** $R^2$ and adjusted $R^2$ values for the multiple linear regression model containing all 88 variables for each cloud macrophysical property.

| Cloud macrophysical property | $R^2$ | Adjusted $R^2$ |
|---|---|---|
| Line fit $\lambda$ | 0.56 | 0.19 |
| Direct power law fit $\lambda$ | 0.58 | 0.21 |
| Fractal dimension | 0.52 | 0.10 |
| Mean cloud top height | 0.85 | 0.72 |
| Cloud fraction | 0.74 | 0.52 |

**Table 4.** Top six meteorological variables that explain most of the variation in the observed mean cloud top height.

| Meteorological variable | Pressure level | Change in $R^2$ |
|---|---|---|
| TCWV | | $-0.14$ |
| EIS | 1000–700 hPa | $-0.12$ |
| LTS | 1000–700 hPa | $-0.11$ |
| Wind speed difference | 1000–975 hPa | $-0.02$ |
| Wind speed difference | 1000–950 hPa | $-0.02$ |
| Vertical velocity | 925 hPa | $-0.02$ |

full regression model using backward elimination (Faraway, 2014) by examining the $p$ values. Starting with the multiple regression model with 88 variables, the variable with the greatest $p$ value (the least significant) was removed. A model was then fit again and the variable with the greatest $p$ value was again removed. This was repeated until all $p$ values were significant at the 0.05 level. After this process, the remaining variables in the model were ranked by the change in $R^2$ when that variable is removed. By doing so, the variables are ranked according to which ones explain most of the variation in the observed macrophysical property. Note that the results do not differ when the remaining variables were instead ranked by the change in the adjusted $R^2$ values.

The final multiple linear regression model for mean cloud top height contains 26 variables, with $R^2 = 0.80$ and adjusted $R^2 = 0.77$. This model has a root mean squared error (RMSE) of 127 m using the leave-one-out cross-validation (LOOCV) technique. Table 4 shows the top six ranked variables by the change in $R^2$ from 0.80 when that variable is removed from the regression model. TCWV, EIS, LTS,

wind speed difference, and vertical velocity are seen to rank highest. These results are consistent with other observational studies that have shown higher TCWV, or total precipitable water, can lead to higher cloud tops in boundary layer clouds (e.g., Forsythe et al., 2012). The stability terms of EIS and LTS are also important as higher values would reduce entrainment drying and warming, moistening the planetary boundary layer and allowing for deeper and more low stratiform clouds (Wood and Bretherton, 2006). Wind speed difference is also important as wind shear can tilt deeper cumulus clouds, limit vertical cloud development, and enhance evaporation at cloud tops (Neggers et al., 2003; Yamaguchi et al., 2019; Helfer et al., 2020), while vertical velocity or subsidence near the cloud top can limit the deepening of marine boundary layer clouds, such as stratocumulus, stratus, and cumulus, due to the pushing down of the top of the marine boundary layer (Myers and Norris, 2013).

The final multiple linear regression model for cloud fraction contains 24 variables, with $R^2 = 0.59$ and adjusted $R^2 = 0.53$. This model has RMSE of 0.068 using the LOOCV

**Table 5.** Top six meteorological variables that explain most of the variation in the observed cloud fraction.

| Meteorological variable | Pressure level | Change in $R^2$ |
|---|---|---|
| Wind speed | 975 hPa | −0.08 |
| Relative humidity | 900 hPa | −0.08 |
| LTS | 1000–900 hPa | −0.08 |
| Wind speed | 925 hPa | −0.06 |
| EIS | 1000–700 hPa | −0.05 |
| LTS | 1000–700 hPa | −0.05 |

technique. Table 5 shows the top six ranked variables by the change in $R^2$ from 0.59 when that variable is removed from the regression model. From the table, wind near the surface is the variable that has the biggest impact on cloud fraction. This agrees with previous modeling and observational studies that have shown how surface wind speed can increase surface fluxes of moisture and heat and deepen the boundary layer, allowing for deeper and larger clouds (Nuijens and Stevens, 2012; Brueck et al., 2015). Naud et al. (2023) also found that the 10 m winds are the dominant cloud-controlling factor for shallow cumulus regions. Relative humidity above the cloud is also important for cloud entrainment and lifetime (Eastman and Wood, 2018). Notice also that the standard EIS and LTS terms calculated between the 1000 and 700 hPa levels are lower in the ranking. This is in agreement with Cutler et al. (2022), who recently showed using surface-based and satellite cloud data, along with reanalysis data, that the correlation between LTS and EIS on cumulus cloud cover is low compared to the correlation of LTS and EIS with stratocumulus cloud cover. Lewis et al. (2023) also recently showed how EIS is not the most important variable for low-cloud cover in the trade cumulus regions. This is an important finding because climate models use these stability terms to parameterize low-cloud cover (Neale et al., 2010) and in climate sensitivity studies to study low-cloud feedback (Brient and Schneider, 2016; McCoy et al., 2017; Myers et al., 2021; Sherwood et al., 2020).

Finally, McFarquhar et al. (2004) and Dey et al. (2011) have also investigated how cloud amount and cloud top height can vary with aerosol amount over the Indian Ocean. Given that the CAMP²Ex mission was held in the Philippines region due to its complex aerosol environment (Reid et al., 2023), the variability in the macrophysical properties with mean aerosol optical depth (AOD) retrieved by MISR for each ASTER scene during the mission was also investigated by examining each of the macrophysical properties as a function of AOD and by grouping scenes based on their AOD into "clean" and "polluted" (not shown here). There was not much variability seen in the cloud macrophysical properties (coefficient of variations ranging from 0.01 to 0.3 for the different properties) as there was not much variability in the AOD observed in the region (standard deviation of

0.10) for the observed ASTER scenes. Most of the ASTER scenes were collected under fairly pristine conditions, with only 7 % of the 550 nm AODs retrieved from MISR having values greater than 0.3. This is in contrast to other studies that have investigated the impact of aerosol loading on the cloud macrophysics. For example, in the study by Dey et al. (2011), which also used ASTER and MISR data, about half of the aerosol retrievals had 550 nm AODs > 0.3. They also had a much larger sample size. Thus, the generality of McFarquhar et al.'s (2004) finding that the cloud macrophysical properties depend on aerosol properties could not be well tested here, given the narrow range in observed AOD over the CAMP²Ex region.

## 6 Conclusions

Cloud macrophysical properties place strong controls on local- to global-scale radiative and latent heat budgets. The macrophysical properties of oceanic shallow cumulus clouds are challenging to model and observe owing to their small sizes and remote locations despite being a commonly occurring cloud type over the oceans, particularly over the tropics. The representation of tropical shallow cumuli in climate models continues to be a major source of uncertainty in cloud feedback. This, in part, calls for improved observations of these clouds and their relationship to meteorological variables to help improve and evaluate their representation in models. This is possible with existing satellite instrument missions focused on land studies, such as ASTER, but such instruments are rarely tasked to collect data over oceans with sampling and settings that are appropriate for deriving the macrophysical properties of clouds. One such occurrence took place as part of the CAMP²Ex field campaign, where the ASTER instrument on board Terra was tasked to collect high-resolution imagery over the ocean waters surrounding the Philippines.

In this study, the macrophysical properties of 2 181 059 cumulus clouds over the tropical western Pacific were examined using 170 ASTER scenes collected from August to October 2019 during the conduct of the CAMP²Ex field campaign. An average cloud fraction of $0.115 \pm 0.14$ was retrieved, with half of that fraction contributed by clouds less than $1.6 \pm 0.1$ km in area-equivalent diameter. Around 80 % of the individual scenes had a cloud fraction less than 0.2. The cloud size distribution follows a power law form, with an exponent of 2.93 ($R = 0.99$) using the line fit method and 2.16 ($R = 0.99$) using the direct power law fit method. An area–perimeter power law was also observed with a dimension of 1.25 ($R = 0.98$), indicating cumulus clouds of smooth shapes. More than 75 % of the clouds were found to have a nearest neighbor within 10 times their area-equivalent radius. After correcting for water vapor absorption that led to $\sim 200$ to 900 m bias in cloud top height (CTH) per scene, the resulting peak frequency in ASTER-derived CTH occurred at

750 m – consistent with MISR and HSRL-2 CTHs. A remaining uncertainty in CTH due to sounding choice was found to be ±160 m. With a mean lifting condensation level (LCL) of 466 ± 89 m for the CAMP²Ex period (Miller et al., 2023), a mode CTH of 750 ± 160 m, and a mode in the cloud fraction distribution occurring in the 400 to 500 m bin, the cloud aspect ratio (cloud depth to width) for this mode is 0.6 ± 0.4. MODIS CTHs were also found to peak in the lowest altitude bin (0 to 250 m) due to the subpixel (1 km MODIS) nature of these clouds. MODIS and MISR standard cloud fraction estimates were also found to have large, positive biases (0.19 and 0.49, respectively) because of the subpixel nature of these clouds, with biases that are consistent with Zhao and Di Girolamo (2006). The newer resolution-corrected cloud fraction product offered by MISR had a small positive bias of 0.02, which is consistent with the expectation of the algorithm (Jones et al., 2012), slightly better than other validation exercises (Dutta et al., 2020), and very close to the 0.014 value of uncertainty in our estimate of the ASTER cloud fraction. Any remaining uncertainty in the macrophysical properties owing to subpixel clouds in 15 m ASTER imagery is expected to be exceedingly small (Dey et al., 2008) relative to the uncertainties reported above.

Similarities and differences were found when results were compared to the previous studies shown in Table 1. Differences between the studies in the table may be due to the different times, location, sampling issues, domain size, and spatial resolution. Given the effect of scale on the statistics, it may be best to compare amongst studies that have used ASTER data (Zhao and Di Girolamo, 2007; Mieslinger et al., 2019). When comparing to these studies, the macrophysical properties of cumulus clouds from the CAMP²Ex region are very similar. A noticeable difference was the higher total cloud fraction in the CAMP²Ex region by around 0.03. The similarities in cloud macrophysics amongst these ASTER studies is remarkable given that they were done in very different parts of the world, although all tropical. We also looked at the 214 scenes of cumulus clouds over the tropical Indian Ocean described in Jones et al. (2012) and again found very similar cloud macrophysical properties (not shown here). We therefore conclude that the properties of tropical trade wind cumulus shown here appear to be indicative of the properties of trade wind cumuli across large swaths of tropical oceanic regions. While aerosol conditions in the environment may impact the macrophysics of shallow oceanic clouds as shown in McFarquhar et al. (2004) and Dey et al. (2011) over the Indian Ocean, the narrow range observed in the AOD for the specific ASTER scenes led to no significant relationship between the AOD and the macrophysical properties.

The relationship between the observed cloud macrophysical properties for each ASTER scene and the meteorological conditions was also investigated. While this has been done using ASTER data in other regions containing oceanic cumuli by Mieslinger et al. (2019) using the average cloud field parameters as a function of single-binned meteorological parameters, we had concerns that artificial relationships could be created between uncorrelated data depending on how the data are binned (Wainer et al., 2006; Rusakov, 2023). To avoid this issue, we used multiple linear regression for each macrophysical property, with the full model having 88 variables. Less than 30 % of the variation in the cloud size distribution parameters (line fit $\lambda$ and direct power law fit $\lambda$) and fractal dimension was explained by all 88 variables. However, for the cloud fraction and mean cloud top height, more than 50 % of their variation can be explained by all 88 variables. Variable selection was performed by backward elimination, reducing the number of variables to 26 for mean cloud top height and to 24 for cloud fraction. It was found that TCWV, LTS, and EIS are the variables that contributed the most to the variation in mean cloud top height, while wind speed near the surface, relative humidity near the cloud, and LTS calculated between 1000 and 900 hPa contribute most to the variation in cloud fraction. Mieslinger et al. (2019) also found that surface wind speed is a huge controlling factor for cloud fraction, along with the standard LTS term, calculated between 1000 and 700 hPa. Interestingly, in our study, we find that the standard LTS and EIS terms, calculated between 1000 and 700 hPa, contribute relatively less to the variation in cumulus cloud fraction, in agreement with Cutler et al. (2022) and Lewis et al. (2023). This is important given that these terms are used in climate models to parameterize overall low-cloud cover (Neale et al., 2010) and in climate sensitivity studies to study low-cloud feedback (Brient and Schneider, 2016; McCoy et al., 2017; Myers et al., 2021; Sherwood et al., 2020). Although not discussed here due to the low adjusted $R^2$ values of the model, similar to cloud fraction, relative humidity (at 1000, 975, and 900 hPa) and wind speed (at 925 and 900 hPa) are the top variables that explain most of the variation in the observed cloud size distribution parameter (line fit $\lambda$) for each ASTER scene. Note again that there is not much confidence in these relationships, however, because only around 30 % of the variation in the line fit $\lambda$ parameter can be explained by the reduced model. We further note that given that the statistics of the macrophysical properties of cumulus clouds are influenced by the domain size, the observed relationships of the cloud macrophysical properties to the meteorological conditions may also change with domain size.

While this study does show some relationship of cloud macrophysical properties to the meteorology, we do not intend to predict or imply how cloud macrophysical properties change with varying meteorology. We explicitly note that the results we presented do not imply any causality. This study simply provides the statistics on the macrophysical properties and shows that meteorology can explain some of the observed variation in the CAMP²Ex region, which has important applications for model evaluation. It should also be noted that any smaller-scale meteorological variations, which are not captured due to the coarse resolution of the ERA5 reanalysis, might be able to explain more of the variation in the

observed properties if such higher-resolution data were available. While the multiple linear regression model is used, this does not imply that the relationships between cloud macrophysical properties and the meteorology are linear. Still, as seen in our results, multiple linear regression is a useful tool to help explain the relationship between the observed properties and the meteorology, showing the relative importance of TCWV to the variations in mean cloud top height and surface wind speed to the variation in cloud fraction.

**Data availability.** The soundings obtained during the CAMP²Ex mission were obtained from https://www-air.larc.nasa.gov/cgi-bin/ArcView/camp2ex (van Diedenhoven et al., 2022), while the standard synoptic soundings acquired over the Philippines were obtained from the University of Wyoming website at https://weather.uwyo.edu/upperair/sounding.html (Department of Atmospheric Science, 2022). The ASTER L1T data were obtained through the NASA Land Processes Distributed Active Archive Center (LP DAAC) data pool at https://lpdaac.usgs.gov/products/ast_l1tv003/ (NASA LP DAAC, 2015). The MISR Level 2 TOA/-Cloud Classifiers product version 3 data were obtained from the NASA Langley Research Center at https://opendap.larc.nasa.gov/opendap/MISR/MIL2TCCL.003/ (NASA LARC, 2008). The MISR Level 2 Cloud product for cloud top height data were obtained from the NASA Langley Research Center at https://opendap.larc.nasa.gov/opendap/MISR/MIL2TCSP.001/ (NASA LARC, 2012). The MODIS Level 2 data CE2, namely MOD35 (Ackerman and Frey, 2015, https://doi.org/10.5067/MODIS/MOD35_L2.006) and MOD06 (Platnick et al., 2015, https://doi.org/10.5067/MODIS/MOD06_L2.061 TS2), were obtained from the Level 1 and Atmosphere Archive and Distribution System of NASA Goddard Space Flight Center at https://ladsweb.modaps.eosdis.nasa.gov/archive/allData/61/ (last access: TS3). The ERA5 reanalysis hourly data at pressure levels were obtained through the Copernicus Climate Change Service Climate Data Store at https://doi.org/10.24381/cds.bd0915c6 (Hersbach et al., 2018a), while the hourly data at single levels were obtained at https://doi.org/10.24381/cds.adbb2d47 (Hersbach et al., 2018b).

**Supplement.** The supplement related to this article is available online at: https://doi.org/10.5194/acp-24-1-2024-supplement.

**Author contributions.** MVDV, LDG, and GZ conceived the study design and analysis. MVDV analyzed data with inputs from LDG, GZ, RMR, SWN, and GMM. LDG acquired funding. MVDV wrote the paper with reviews from co-authors.

**Competing interests.** At least one of the (co-)authors is a member of the editorial board of *Atmospheric Chemistry and Physics*. The peer-review process was guided by an independent editor, and the authors also have no other competing interests to declare.

**Disclaimer.** Publisher's note: Copernicus Publications remains neutral with regard to jurisdictional claims made in the text, published maps, institutional affiliations, or any other geographical representation in this paper. While Copernicus Publications makes every effort to include appropriate place names, the final responsibility lies with the authors.

**Special issue statement.** This article is part of the special issue "Cloud, Aerosol and Monsoon Processes Philippines Experiment (CAMP2Ex) (ACP/AMT inter-journal SI)". It is not associated with a conference.

**Acknowledgements.** The authors would like to acknowledge Dongwei Fu for providing the histogram data for the CTH distribution from the HSRL-2 instrument used in Fu et al. (2022). We also thank Sonia Lasher-Trapp for her valuable comments and discussion on this paper.

**Financial support.** This research has been supported by the National Aeronautics and Space Administration through grant nos. 80NCCS18K0144 and 80NNC21K1449 and was partially supported by the MISR project through the Jet Propulsion Laboratory of the California Institute of Technology. Greg M. McFarquhar was supported by the Cooperative Institute for Severe and High-Impact Weather Research and Operations (CIWRO) and the NASA AOS program through grant no. 80NSSC23M0083. CE3

**Review statement.** This paper was edited by Yi Huang and reviewed by two anonymous referees.

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

## Remarks from the language copy-editor

CE1    For consistency, this was changed as well.

CE2    Please verify.

CE3    Please verify this section. Note that it is a house standard to avoid language like "acknowledged" in this section.

## Remarks from the typesetter

TS1    Please provide a short explanation regarding the addition of $f$ in the equation above. This change will have to be approved by the editor.

TS2    Please confirm addition of DOIs.

TS3    Please provide date of last access.

TS4    Please check URL.