# Peer review of "Observations of the macrophysical properties of cumulus cloud fields over the tropical western Pacific and their connection to meteorological variables"

_EGUsphere, 2023_

## Author Comment (AC1)

We would like to thank the editor for handling the review process for this manuscript. We are also grateful to the reviewers for their insightful comments. We were able to incorporate changes to reflect all of the suggestions of the reviewers and have highlighted these changes within the manuscript. The line numbers in our responses refer to the line numbers in the revised manuscript.

**Response to reviewer 1:**

The manuscript presents an analysis of the macrophysical properties of cumulus cloud fields over the tropical western Pacific using high-resolution ASTER satellite data during the CAMP2Ex mission. This study also reveals the correlations between these properties and various meteorological variables. The authors show that the average cloud fraction is notably contributed by smaller clouds and that the variation in mean cloud top height is significantly affected by total column water vapor and LTS. Overall, this paper is well-written with clear scientific merit. However, the uncertainties and causality need to be strengthened. With this, I would recommend the publication of this manuscript with the following major revisions.

Specific comments:
1. This work can strengthen the discussions of uncertainties associated with deriving cloud macrophysical properties from ASTER data. This discussion would benefit from a more comprehensive examination of potential biases and errors that could impact the interpretation of results, especially when using high-resolution satellite imagery.

Our original manuscript did provide an examination of the uncertainties in our results, including the much-reduced uncertainties in using high-resolution (15 m) ASTER over more typical moderate resolution sensors. Granted, it was terse for some variables because the uncertainties in those variables were comprehensively examined in other works that we cited (e.g., Wielicki and Welch 1986; Di Girolamo and Davies 1997; Zhao and Di Girolamo 2006 and 2007; Dey et al. 2008). For variables where we could not rely on other works, our uncertainty analysis was comprehensive. For example, most of Section 4.5 that reports on cloud top heights in the original manuscript was about assessing uncertainty in these heights.

We very much appreciate the reviewer's concerns on this, so we added some clarifying remarks on uncertainties throughout, and added a major revision to the paragraph in the conclusion section that summarizes our findings to now also include their uncertainties as quantified and discussed throughout Section 4. This paragraph now reads as follows in lines 583-602 in the conclusion section:

"In this study, the macrophysical properties of 2,181,059 cumulus clouds over the tropical western Pacific were examined using 170 ASTER scenes collected from August to October 2019 during the conduct of the CAMP$^2$Ex field campaign. An average cloud fraction of $0.115 \pm 0.014$ was retrieved, with half of that fraction contributed by clouds less than $1.6 \pm 0.1$ km in area equivalent diameter. Around 80 % of the individual scenes had a cloud fraction less than 0.2. The cloud size distribution follows a power law form, with an exponent of 2.93 ($R = 0.99$) using the line-fit method and 2.16 ($R = 0.99$) using the direct power-law fit method. An area-perimeter

power law was also observed with a dimension of 1.25 ($R$ = 0.98), indicating cumulus clouds of smooth shapes. More than 75 % of the clouds were found to have a nearest neighbor within 10 times their area-equivalent radius. After correcting for water vapor absorption that led to ~200 to 900 m bias in cloud top height (CTH) per scene, the resulting peak frequency in ASTER-derived CTH occurred at 750 m – consistent with MISR and HSRL-2 CTHs. A remaining uncertainty in CTH due to sounding choice was found to be ±160 m. With a mean lifting condensation level (LCL) of 466 ± 89 m for the CAMP²Ex period (Miller et al., 2023), a mode CTH of 750 ± 160 m, and a mode in the cloud fraction distribution occurring in the 400 to 500 m bin, the cloud aspect ratio (cloud depth to width) for this mode is 0.6 ± 0.4. MODIS CTHs were also found to peak in the lowest altitude bin (0 to 250 m) due to the subpixel (1 km MODIS) nature of these clouds. MODIS and MISR standard cloud fraction estimates were also found to have large, positive biases (0.19 and 0.49, respectively) because of the sub-pixel nature of these clouds, with biases that are consistent with Zhao and Di Girolamo (2006). The newer "resolution-corrected" cloud fraction product offered by MISR had a small positive bias of 0.02, which is consistent with the expectation of the algorithm (Jones et al., 2012), slightly better than other validation exercises (Dutta et al., 2020), and very close to the 0.014 value of uncertainty in our estimate of the ASTER cloud fraction. Any remaining uncertainty in the macrophysical properties owing to sub-pixel clouds in 15-m ASTER imagery is expected to be exceedingly small (Dey et al. 2008) relative to the uncertainties reported above."

We also address the reviewer's concerns by further clarifying how we arrived at the uncertainties and repeating some key points where appropriate throughout Section 4, specifically:

Section 4.1, lines 217-218:
"Note again these results are insensitive to small changes in the cloud masking threshold used but can differ with other studies due to domain size and spatial resolution."

Section 4.2, lines 233-235:
"The average cloud fraction from all 170 scenes is 0.115 ± 0.014. This uncertainty comes from half the fraction of cloud edge pixels (Di Girolamo and Davies, 1997), which is 0.027."

Section 4.3, lines 286-287:
"Note that the fractal dimension $d$ will also be insensitive to small perturbations in the cloud threshold as Cahalan (1991) showed only a 0.1 increase in dimension with around a 300% increase in threshold."

Section 4.5, lines 326-328:
"Note that small perturbations to the choice of the cloud detection threshold will only impact the number of 90-m pixels used to retrieve CTH, but not the values of CTH (Zhao and Di Girolamo, 2007)."

2. The manuscript utilizes R-squared values to infer meteorological controls over cloud properties. However, there is an imitation of correlations as an indicator, particularly if there are other confounding factors, such as synoptic patterns, that can affect both meteorology and cloud properties simultaneously. The correlation does not clearly reveal the causality.

We agree that correlation does not imply causation and do not claim our results imply causality in our paper. This was one of the reasons why we decided to do multiple linear regression as opposed to a single linear regression for each meteorological parameter as was done in previous studies (e.g., Mieslinger et al. 2019; see lines 449-456). This was also why we did not rank the variables by their coefficients, but by how removing them from the model can change the R-squared value. In doing so, we see which variables explain most of the variation in the cloud property. We then turn to published modelling studies (e.g. Neggers et al., 2003; Yamaguchi et al., 2019; Helfer et al., 2020 in lines 528-530 and Nuijens and Stevens, 2012 in lines 539-541) that are capable of examining causality to support the rankings found in our study.

We addressed the reviewer's concern by explicitly taking note of this in the conclusions (lines 693-694):

"We explicitly note that the results we presented do not imply any causality."

3. While the impact of meteorological parameters on cloud top height and fraction is explored, this work can discuss more on the key parameters controlling cloud size, such as cloud equivalent diameter. Including an analysis of the impacts of factors on the cloud size could provide a more comprehensive understanding of the cumulus clouds.

We agree that the relationship between the meteorological parameters and the cloud size distribution is an important aspect to explore. We did not fully discuss this more in the manuscript as the adjusted R-squared values in the multiple linear regression models for the cloud size distribution parameters using all 88 variables were only around 0.3. After model reduction, the R-squared values and adjusted R-squared values remained around 0.3. This may be due to the low variability seen in these properties among all the scenes (see the supplementary material). This discussion can be found in lines 503-507.

We now clarify this point by adding a few sentences on our results for the cloud size distribution parameter (line-fit $\lambda$) in the conclusions (lines 683-687):

"Although not discussed here due to the low adjusted $R^2$ values of the model, similar to cloud fraction, relative humidity (at 1000, 975, and 900 hPa) and wind speed (at 925 and 900 hPa) are the top variables that explain most of the variation in the observed cloud size distribution parameter (line-fit $\lambda$) for each ASTER scene. Note again that there is not much confidence in these relationships, however, because only around 30 % of the variation in the line-fit $\lambda$ parameter can be explained by the reduced model."

4. The study could be strengthened by including analyses of planetary boundary layer (PBL)-related parameters, like PBL height and PBL stability, given their known influence on cumulus cloud modulation. This inclusion would provide a more in-depth perspective on the interactions between the PBL and cloud properties.

We have taken the reviewer's suggestion, which may be based on studies such as Stevens et al. (2007) and Kubar et al. (2015) that have shown the relationship between PBL and cloud fraction and depth for stratocumulus clouds. We downloaded ERA5 reanalysis data for the boundary layer height (BLH) and included this into the multiple linear regression model. Following our methodology, after model reduction, the BLH was not found to be significant for all cloud properties and was eliminated by p-value. The table below shows the p-values for BLH in the full multiple linear regression model (using all 89 variables) for each cloud property.

| Cloud Macrophysical Property | p-value for BLH |
|---|---|
| Line-Fit $\lambda$ | 0.838 |
| Direct Power-Law Fit $\lambda$ | 0.233 |
| Fractal Dimension | 0.625 |
| Mean Cloud Top Height | 0.171 |
| Cloud Fraction | 0.229 |

As shown above, all p-values for BLH are larger than 0.05 and were thus not part of the final reduced models. This may likely be due to the decoupling of the PBL in the trade cumulus regime, leading to relatively shallow and small cumuli with CTHs well below the PBL top as found in other studies (Karlsson et al., 2010; Kubar et al., 2020). As such, we have decided not to include the variable. Instead, we have taken note of this and written this in the revised manuscript in lines 468-472 as:

"Note that the ERA5 boundary layer height (BLH) variable was also examined and was found to be insignificant based on p-value for all the observed macrophysical properties. Therefore, it was not included in the final list of variables shown in Table 2. The low ranking may be due to the decoupling of the PBL in the trade cumulus regime, leading to relatively shallow and small cumuli with CTHs well below the PBL top as found in other studies (Karlsson et al., 2010; Kubar et al., 2020)."

Karlsson, J., Svensson, G., Cardoso, S., Teixeira, J., and Paradise, S.: Subtropical cloud-regime transitions: Boundary layer depth and cloud-top height evolution in models and observations, J. Appl. Meteorol. Clim., 49, 1845-1858, doi: 10.1175/2010JAMC2338.1, 2010.

Kubar, T. L., Stephens, G. L., Lebsock, M., Larson, V. E., and Bogenschutz, P. A.: Regional assessments of low clouds against large-scale stability in CAM5 and CAM-CLUBB using MODIS and ERA-Interim reanalysis data, J. Climate, 28(4), doi: 10.1175/jcli-d-14-00184.1, 1685–1706, 2015.

Kubar, T. L., Xie, F., Ao, C. O., and Adhikari, L.: An assessment of PBL heights and low cloud profiles in CAM5 and CAM5-CLUBB over the Southeast Pacific using satellite observations, Geophys. Res. Lett., 47, e2019GL084498, doi: 10.1029/2019GL084498, 2020.

Stevens, B., Beljaars, A., Bordoni, S., Holloway, C., Köhler, M., Krueger, S., and Zhang, Y.: On the structure of the lower troposphere in the summertime stratocumulus regime of the northeast pacific, Mon. Weather Rev., 135, 985–1005, doi: 10.1175/mwr3427.1, 2007.

**Response to reviewer 2:**

Overall, I think the manuscript is well-written and is worthy of rapid publication. Very minor suggestions are provided below.

A few references could be added, for a more complete literature review.

Lines 35-36:
You discuss the morphology of clouds, but I think you need other references in addition to Tobin et al., (2013). This issue and the issue of cloud heterogeneity are discussed in many recent papers. The following references should be considered:

Rampal, N., & Davies, R. (2020). On the factors that determine boundary layer albedo. *Journal of Geophysical Research: Atmospheres*, *125*(15), e2019JD032244.

Lang, F., Ackermann, L., Huang, Y., Truong, S. C., Siems, S. T., & Manton, M. J. (2022). A climatology of open and closed mesoscale cellular convection over the Southern Ocean derived from Himawari-8 observations. *Atmospheric Chemistry and Physics*, *22*(3), 2135-2152.

Lang, F., Siems, S. T., Huang, Y., Alinejadtabrizi, T., & Ackermann, L. (2024). On the relationship between mesoscale cellular convection and meteorological forcing: comparing the Southern Ocean against the North Pacific. *Atmospheric Chemistry and Physics*, *24*(2), 1451-1466.

Goren, T., Sourdeval, O., Kretzschmar, J., & Quaas, J. (2023). Spatial Aggregation of Satellite Observations Leads to an Overestimation of the Radiative Forcing Due To Aerosol-Cloud Interactions. *Geophysical Research Letters*, *50*(18), e2023GL105282.

We have included these references in lines 37-38 as:
"Other studies (Rampal and Davies, 2020; Goren et al., 2023, Lang et al., 2024) have further shown how cloud morphology and cloud heterogeneity can impact the measured radiative field."

Lines 80 - 85, could use more recent references such as:
Lewis, H., Bellon, G., & Dinh, T. (2023). Upstream Large-Scale Control of Subtropical Low-Cloud Climatology. *Journal of Climate*, *36*(10), 3289-3303.

We have included this reference in lines 546-547 instead, where it seems more fitting:
"Lewis et al. (2023) also recently showed how EIS is not the most important variable for low cloud cover in the trade-cumulus regions."

However, we have also included more recent references to lines 83-87 as follows:
"Some of these findings, for example, indicate that the lower-tropospheric stability (LTS), estimated inversion strength (EIS; Wood and Bretherton, 2006; McCoy et al., 2017), reduced subsidence (Myers and Norris, 2013; Blossey et al., 2013; van der Dussen et al., 2016), sea surface temperature (Qu et al., 2015; Stein et al., 2017; Geiss et al., 2020; McCoy et al., 2017), and surface wind speed (Bretherton et al., 2013) all can have an impact on cloud cover and cloud top height."

I encourage the authors to add other more recent references too.

We have added other recent references to the following:

Lines 528-530: "Wind speed difference is also important as wind shear can tilt deeper cumulus clouds, limit vertical cloud development, and enhance evaporation at cloud tops (Neggers et al., 2003; Yamaguchi et al., 2019; Helfer et al., 2020),…"

Lines 541-542: "Naud et al. (2023) also found that the 10-m winds are the dominant cloud controlling factor for shallow cumulus regions."

We have also ensured to include the aforementioned references in the 'References' section of our manuscript for complete bibliographic integrity.